# Oriented Deep Eutectic Solvents as Efficient Approach for Selective Extraction of Bioactive Saponins from Husks of *Xanthoceras sorbifolia* Bunge

**DOI:** 10.3390/antiox11040736

**Published:** 2022-04-08

**Authors:** Jinteng Cao, Guangwei Wu, Lei Wang, Fuliang Cao, Yan Jiang, Linguo Zhao

**Affiliations:** 1Jiangsu Co-Innovation Center of Efficient Processing and Utilization of Forest Resources, Nanjing Forestry University, Nanjing 210037, China; Jintengcao@njfu.edu.cn (J.C.); gweiwu@njfu.edu.cn (G.W.); wl0308@njfu.edu.cn (L.W.); 2College of Chemical Engineering, Nanjing Forestry University, Nanjing 210037, China; 3Co-Innovation Center for Sustainable Forestry in Southern China, Nanjing Forestry University, Nanjing 210037, China; flcao@njfu.edu.cn

**Keywords:** deep eutectic solvents, *Xanthoceras sorbifolia* Bunge, saponin, response surface methodology, by-product

## Abstract

The husks of *Xanthoceras sorbifolia* Bunge (*X. sorbifolia*), as by-products of industrial production, have brought a severe burden to the environment and caused an enormous waste of resources. Bioactive triterpenoid saponins are rich in the husks. To reuse the husks and gain high-quality saponin products, saponin-oriented deep eutectic solvents (DESs), as an efficient and selective extraction strategy from *X. sorbifolia* husks, were designed for the first time. The enhancement of the extraction rate was investigated by screening solvents from acidic DESs and response surface methodology (RSM) optimization. As a result, the tetrapropylammonium bromide-lactic acid (TPMBr-La) was the most efficient DESs, with an extraction efficiency of up to 135% higher than 70% ethanol. A maximum extraction rate of 72.11 ± 0.61 mg Re/g dw was obtained under the optimized parameters. Scanning electron microscope graphs revealed that damage to the microstructure caused by DESs enhanced the extraction efficiency. Moreover, the recovery of total saponins with D101 macroporous resin was consistent with the pseudo-second-order kinetic model. Seven saponins were also identified by HPLC-MS analysis. Finally, TPMBr-La extracts exhibited 92.30 ± 1.10% DPPH radical scavenging rate at 100 μg/mL, and 92.20 ± 0.30% ABTS radical scavenging rate at 1200 μg/mL. Our current research proposes a selective and high-efficiency substitute for the extraction of saponins and might contribute to further DESs application in the recycling of by-products.

## 1. Introduction

*Xanthoceras sorbifolia* Bunge. (*X. sorbifolia*), belonging to the Sapindaceae family, is a native woody oil crop and has a planting history of more than 1000 years in China [1]. Due to its strong survival ability and adaptability [2], it is very suitable for planting in different climatic conditions and is also an ecological tree species and windproof against the sand. In recent decades, *X. sorbifolia* has attracted significant attention due to its promising potential for the production of high-quality biodiesel [3], pharmaceutical [4,5,6] and for ecological purposes [7] worldwide. As of 1998, the planting area of *X. sorbifolia* has been expanded to 800,000 hm^2^ [8]. At present, the development of biodiesel made from the seeds of the *X. sorbifolia* faces two major problems: the high number of low value-added wastes, such as fruit husks, produced in the production process, and the high cost of production. The mass of *X. sorbifolia* husks accounts for about 50% of the overall fruit mass [9]. The annual output of 100,000 tons of biodiesel requires the simultaneous consumption of 300,000 tons of *X. sorbifolia* seeds and the production of 300,000 tons of husks [10]. A high number of husks are discarded as waste or low-value-added products after biodiesel production every year, which are not effectively utilized, resulting in greater environmental pressure and the waste of resources. Research into the application of *X. sorbifolia* husks has focused on activated carbon [11] and feed. Due to the high saponins content in the husks, they are not conducive to the further development of feed. Therefore, the high-value utilization and deep processing of *X. sorbifolia* husks resources is not only a critical aspect for the alleviation of pressure on the environment and the reduction in production costs, but is also a breakthrough in the application of *X. sorbifolia* husks.

According to the research, barrigenol-like triterpenoid saponins are responsible for a range of pharmacological bioactivities, such as anti-tumor, anti-bacterial, anti-Alzheimer’s disease, anti-neuroinflammatory, and antioxidant properties [12,13,14]. Some barrigenol-like triterpenoid derivatives have also been applied to the treatment of encephaloedema [15]. *X. sorbifolia* husks are known for their abundant barrigenol-like triterpenoid saponins [16] and flavonoids; thus, it is clear that the *X. sorbifolia* husks have great utilization value and potential to develop high-value-added products, especially in the pharmaceutical industry. In recent years, some studies have focused on extracting saponins and other bioactive components from *X. sorbifolia* husks with conventional organic solvents, including ethanol, ethyl acetate, n-butanol, and ether [17]. However, traditional methods for the extraction of saponins from *X. sorbifolia* husks can result in low product purity, solvents residues, and high solvent consumption, which may cause serious environmental pollution and is not conducive to subsequent production. Eco-friendly and efficient methods have received extensive attention in recent years [18]. Considering the promising application of saponins from *X. sorbifolia* and the improved waste utilization rate, it is of great significance to seek new methods to achieve the high-efficiency extraction of saponins from *X. sorbifolia* husks and provide the extraction process with higher specificity.

Deep eutectic solvents (DESs) have been regarded as eco-friendly solvents that can substitute traditional organic solvents [19]. DESs are usually composed of hydrogen bond acceptors (HBA_s_) and hydrogen bond donors (HBD_S_) [20], with intermolecular hydrogen bonds and van der Waals interactions [21]. DESs have the advantages of being adjustable and biodegradable, with lower toxicity and low costs [22,23]. Tailor-made DESs have been applied to the selective extraction of bioactive components such as saponins, phenolics, and flavonoids. For example, acetic acids acting as HBDs in DESs have, in general, higher extraction efficiency for *Agave sisalana* and *Ziziphus joazeiro* due to the better solubilization of saponins [24]. Besides, the extraction rate of four steroidal saponins from Dioscoreae Nipponicae Rhizoma by natural DESs composed of chloride and malonic acid was higher, and the difference was significant compared to the methanol extraction [25]. Unfortunately, the oriented DESs approach towards extracting saponins from *X. sorbifolia* husks has not yet been investigated.

In the present study, we attempted to develop a selective and efficient approach to the extraction of total saponins from *X. sorbifolia* husks using saponins-oriented DESs for the first time. The difference in extraction efficiency and the possible mechanism were compared and determined. Moreover, extraction conditions were optimized by RSM. In addition, the recovery of saponins was carried out and the adsorption kinetics of macroporous resin was investigated. Finally, the saponins components of the extracts were determined by HPLC-ESI-MS analysis, and the antioxidant activities of the recovered TPMBr-La extracts were evaluated. Our findings provide a new strategy for the extraction of bioactive saponins from *X. sorbifolia* shells and the reuse of industrial by-product resources.

## 2. Materials and Methods

### 2.1. Plant Materials and Reagents

*Xanthoceras sorbifolia* Bunge. (NCBI-txid99658) husks were collected in October 2020 from *Xanthoceras sorbifolia* Bunge. standardized demonstration garden (Anqiu City, Weifang City, Shandong Province, China) and identified by associate professor Yu Wanwen (Co-Innovation Center for the Sustainable Forestry in Southern China, Nanjing Forestry University, Nanjing, China). The husks were dried in an oven at 40 °C to a constant weight, ground to 60 mesh with a pulverizer, and the powders were stored airtight.

All reagents were purchased from Macklin Biochemical Technology (Shanghai, China), unless otherwise specified. Total Antioxidant Capacity Assay Kit was purchased from Beyotime Biotechnology Ltd. (Shanghai, China). Macroporous resins were acquired from Yuanye Biotechnology Ltd. (Shanghai, China). Methyl alcohol (HPLC grade) was purchased from Anhui Tedia High-Purity Solvents Co., Ltd (Anhui, China).

### 2.2. Preparations of DESs

The preparation methods used in this study are reported in the literature [26]. The components of selected DESs were weighed at the 1:2 molar ratio in 50 mL centrifuge tubes and stirred at 90 °C until clear liquids were obtained. Fourteen types of DESs were prepared; their compositions and solvent codes listed in Table 1.

### 2.3. Extraction of Total Saponins and Total Flavonoids

A total of 100 mg of *X. sorbifolia* powder and 2 mL of solvents with 30% water content (DESs, ethanol, and water) were added to a 10-mL centrifugal tube. The mixtures were shaken in a water-bath shaker at 60 °C for 30 min. Then, they were centrifuged at 12,000 rpm for 20 min to take the supernatant for determination.

### 2.4. Microstructure of Plant Material

The microstructure graphs of the *X. sorbifolia* powder treated by different solvents (water, ethanol, and TPMBr-La) were detected under thermal field emission scanning electron microscope JSM-7600F (Japan). The extracts were centrifuged to extract the precipitate and dried at low temperature in an oven. After being covered with gold, the microstructure was observed under a 3000-times magnification.

### 2.5. Optimal DESs Extraction Conditions for Total Saponins

Extraction variables and ranges were determined based on previous single-factor experiments (Appendix A). Box-Behnken design experiment was conducted to optimize the extraction conditions. Extraction time (min), extraction temperature (°C), liquid-solid ratio (mL/g), and water content (%) were selected as independent variables, and the extraction rate of total saponins (mg/g) was used as the response value. Data were analyzed by software Design Expert 10.0.7. The obtained experimental data were consistent with the following second-order polynomial model:(1)$${Y=\beta }_{0}+\overset{4}{\underset{i=1}{\sum }}\beta_{i}x_{i}+\overset{4}{\underset{i\;<\;j=1}{\sum }}\beta_{\mathrm{ij}}x_{i}x_{j}+\overset{4}{\underset{i=1}{\sum }}\beta_{\mathrm{ii}}x_{i}^{2}$$
where Y is the response value, X_i_ and X_j_ are the independent variables, β_0_ is a constant, β_i,_ β_ii_, β_ij_ are coefficients for linear, quadratic, and interaction parameters.

### 2.6. Determination of Total Saponins and Total Flavonoids Content

#### 2.6.1. Determination of Total Saponins Content

The quantitative determination of total saponins was performed by the method previously reported in the literature [27]. Mixed 0.5 mL saponin extract with 0.2 mL 5% vanillin and 0.8 mL perchloric acid solution in a 10-mL test tube. The tubes were kept at 60 °C for 15 min and cooled for 2 min; the sample absorbances were measured at 552 nm. According to the relationship between absorbance value and different concentration of ginsenoside Re, the standard curve equation was acquired as follows: y = 14.043x + 0.0833 (0.2 < x < 1 mg/mL), R^2^ = 0.9997. The extraction rate was represented by mg ginsenoside Re equivalent (mg Re/g dw).

#### 2.6.2. Determination of Total Flavonoids Content

The quantitative determination of total flavonoids was performed by the method previously reported in the literature [28]. A total of 1 mL extracted flavonoid samples were well-mixed with 0.3 mL of 5% NaNO_2_ solution; then, the sample was left for 6 min. Then, 0.3 mL of 10% Al(NO)_3_ solution was added, shook well, and placed in the sample for 6 min. Next, 4 mL of 4% NaOH solution was mixed, and the volume was fixed to 10 mL with 80% ethanol solution. Finally, samples were left at room temperature for 15 min and absorbances were measured at 510 nm. According to the relationship between absorbance value and different rutin concentrations, the standard curve equation was acquired as follows: y = 6.1043x + 0.0492 (0.04 < x < 1.5), R^2^ = 0.9994, indicating that the linearity of rutin was good in the concentration range from 0.04 to 1.5 mg/mL. This method can be used for the quantitative analysis of flavonoids; the extraction rate was represented by mg rutin equivalent (mg rutin /g dw), and calculated as follows:(2)$$\mathrm{Extraction}\;\mathrm{rate}\;(\mathrm{mg}/g)=\frac{\mathrm{mass}\;\mathrm{of}\;\mathrm{the}\;\mathrm{crude}\;\mathrm{saponins}\;\mathrm{or}\;\mathrm{flavonoids}}{\mathrm{mass}\;\mathrm{of}\;\mathrm{the}\;\mathrm{shell}\;\mathrm{powder}}$$

### 2.7. Qualitative Analysis of Recovered Saponins Composition by HPLC-ESI-MS

Qualitative analysis of recovered saponins was performed on an HPLC chromatographic system (Agilent, Palo Alto, CA, USA) coupled with an LTQ-Qrbitrap mass spectrometer (Thermo Fisher Scientific, Waltham, MA, USA). The chromatographic with a ZORBAX SB-Aq C_18_ column (250 × 4.6 mm, 5 µm, Agilent, Palo Alto, CA, USA). The mobile phase were methyl alcohol (C) and 0.01% formic acid water (D); the flow rate was 0.4 mL/min. Gradient elution was conducted as follows: 0–30 min, 50–70% C; 30–60 min, 70–80% C; 60–70 min, 80–50% C. Injection volume was 10 µL and injection temperature was 30 °C. MS parameters were as follows: capillary temperature, 250 °C; auxiliary gas heater temperature, 200 °C; spray voltage, 1.5 kV; scanning range, 300–2000 *m*/*z*.

### 2.8. Recovery of Bioactive Saponins from TPMBr-La Extracts

#### 2.8.1. Screening of Macroporous Resins

The recovery of saponins from DESs extract was carried out by adsorption and desorption with five kinds of macroporous resins, including DM301, NKA-9, D101, HPD100, and HP20. Saponins extracted by TPMBr-La were isolated and purified in accordance with the modified method [29]. A total of 2 g of the treated resin was weighed separately in a 50-mL conical bottle, and 10 mL of saponins solution extracted by TPMBr-La was added, before shaking at 25 °C with a speed of 80 rpm. The supernatant was sampled at regular intervals to detect the saponin content until the adsorption equilibrium was reached, before calculating the adsorption capacity (q_e_) and adsorption rate (Q_ar_). The saturated macroporous resin was washed, and 10 mL of 70% ethanol solution was used as desorption solution, and shaken a shaking table at 25 °C with a speed of 80 rpm to desorption equilibrium, before calculating the desorption rate (Q_dr_).
(3)$$q_{e}=\left(C_{0}-C_{e}\right){\times V}_{a}/W$$
(4)$$Q_{\mathrm{ar}}=\left(C_{0}-C_{e}\right)/C_{0}\times 100\%$$
(5)$$Q_{\mathrm{dr}}{=C}_{d}{\times V}_{d}∕{\;((C}_{0}-C_{e}{)\times V}_{a})\times 100\%$$
where C_0_ and C_e_ are the concentration (mg/mL) of total saponins at the beginning and equilibrium of adsorption process, V_a_ is the loading volume (mL) and W is the weight of macroporous resins (g). C_d_ (mg/mL) represents the concentration of saponins in the desorption eluent and V_d_ is the eluent volume (mL) in desorption experiments.

#### 2.8.2. Study on Adsorption Kinetics of D101 Macroporous Resin

A total of 2 g of treated D101 macroporous resin was weighed into a 50 mL conical flask. Then, a total of 10 mL DESs extract was added, and shaken at 25 °C and 80 rpm for 7 h in a constant-temperature shaker. The supernatant was sampled at every 30 min to detect the saponin content until the adsorption equilibrium was reached. The adsorption rate and mechanism were studied by fitting the intra−particle diffusion, pseudo-first-order, and pseudo-second-order kinetic models. The linear equations of the three models are as follows:(6)Intra-particle-diffusion: q=ki12+C
(7)$$\mathrm{Pseudo}-\mathrm{first}-\mathrm{order}:\;\ln\left(q_{e}-q\right){=-k}_{1}{t+\mathrm{lnq}}_{e}$$
(8)$$\mathrm{Pseudo}-\mathrm{second}-\mathrm{order}:\;\frac{t}{q}=\frac{1}{k_{2}q_{e}^{2}}+\frac{t}{q_{e}}$$
where q_e_ and q are adsorption capacity (mg/g) at the equilibrium and t min; k_i_, k_1_, and k_2_ represent the adsorption rate constants of intra particle diffusion, pseudo-first-order, and pseudo-second-order kinetic models; C is the kinetic constant of intra particle diffusion, mg/g.

### 2.9. Determination of Antioxidant Activity

#### 2.9.1. DPPH Radical Scavenging Assay

DPPH radical scavenging assay was determined by this method with slight modification [30]. The sample was prepared into a series of concentrations with methanol. In brief, 100-µL sample solutions were mixed with the same volume of DPPH solution. The mixture was placed away from light for 30 min; then, the absorbances were read at 517 nm. The DPPH radical scavenging rate of samples was assessed in accordance with the following equation:(9)$$\mathrm{Radical}\;\mathrm{scavenging}\;\mathrm{rate}\;(\%)=\frac{A_{\mathrm{control}}-{\;A}_{\mathrm{sample}}}{A_{\mathrm{control}}}\times 100\%$$

#### 2.9.2. ABTS Radical Scavenging Assay

The ABTS radical scavenging capacity was determined by referring to the instructions of Total Antioxidant Capacity Assay Kit. ABTS could generate ABTS radical cations (ABTS^+^) in the presence of an oxidant. A total of 200 μL ABTS working solution and 10-μL diluted samples were mixed in a clear 96-well microplate. The microplate was allowed to stand for 6 min at room temperature and the absorbance was measured at 734 nm. The ABTS radical scavenging rate of the sample was calculated in accordance with Equation (9).

### 2.10. Statistical Analysis

Except for HPLC-ESI-MS analyses, all experiments were performed in triplicate. Data were expressed as mean ± standard deviation. The statistically significant differences (*p* < 0.05) were conducted by analysis of variance (ANOVA).

## 3. Results and Discussion

### 3.1. Design and Screening of Saponin−Oriented DESs

With the advantages of acidic DESs in saponin extraction [24], we selected lactic acid, malonic acid, glycolic acid, and malic acid as HBDs. Choline chloride, betaine, ethylamine hydrochloride, tetrapropylammonium bromide, and ethylene glycol were chosen as HBAs. To facilitate analysis, fourteen designed DESs were classed into five groups based on HBAs. In contrast with traditional organic solvents, the high viscosity of the DESs affected the mass transfer rate, so 30% water was added to the DESs to ease the viscosity [31]. Meanwhile, considering the difference in the extraction efficiency of DESs for saponins and flavonoids, flavonoids, as an important component in *X. sorbifolia* husks, were also chosen to estimate the specific and efficient DESs for saponins. The extraction rates of the total saponins and total flavonoids with different DESs groups and conventional efficient solvents, i.e., 70% ethanol and water are exhibited in Figure 1. It is indicated that the component of DESs extensively impacted the extraction efficiency for the total saponins and total flavonoids, respectively. As shown in Figure 1a, the Chcl-based DESs group had the highest overall extraction efficiency for saponins, followed by the TPMBr-based, Eacl-based, and Bet-based groups. The Eg-based group showed the lowest extraction efficiency for saponins. The extraction efficiency of most of the acidic DESs for saponins was higher than when using 70% ethanol and water, and this finding is in accordance with the previous report [32]. Among the 14 kinds of DESs, TPMBr-La (67.01 ± 0.57 mg Re/g dw) showed the highest extraction rate for saponins, up to 135% higher than the use of 70% ethanol (28.50 ± 0.02 mg Re/g dw) and 296% higher than the use of water (16.90 ± 0.60 mg Re/g dw). Moreover, Chcl-Ga, Bet-La, Eg-La, and Eacl-La also displayed an efficient extraction rate (60.72 ± 2.53 mg Re/g dw, 54.24 ± 1.99 mg Re/g dw, 51.49 ± 2.53 mg Re/g dw, and 51.14 ± 2.90 mg Re/g dw). Bromine ion can connect the hydroxyl H atom on saponins and the solvent, to reduce the spatial effect around saponins and form more hydrogen bonds [33]. The comprehensive score for TPMBr-La for the total saponins was optimal, which could be attributed to the growth in the number of hydrogen bonds and more stable average noncovalent interaction.

The overall extraction efficiency of the acidic DESs for total flavonoids was obviously lower than that for saponins. In Figure 1b, the Chcl-Ga, Bet-La, and Eg-La (41.99 ± 0.63 mg Rutin/g dw, 33.82 ± 1.22 mg Rutin/g dw, 36.36 ± 0.03 mg Rutin/g dw, respectively) showed a significant extraction efficiency rate for flavonoids. However, TPMBr-La (23.57 ± 3.40 mg Re/g dw) displayed a lower extraction efficiency for flavonoids, suggesting that TPMBr-La is selective for the extraction of saponins. Taken together, TPMBr-La was the best DESs to specificly extract bioactive saponins from *X. sorbifolia* and would be applied in the follow-up experiments.

### 3.2. Preliminary Extraction Mechanism for the Efficiency and Specificity of Saponins Extraction

To better understand why TPMBr-La was the most effective method for the extraction of saponins, the different extraction solvents, including water, 70% ethanol, and TPMBr-La, were investigated. The microstructure graphs of the *X. sorbifolia* powder before and after treatment using the above extraction solvents were recorded using thermal field emission scanning electron microscope. The *X. sorbifolia* powder without extraction displayed a flat and undamaged surface (Figure 2a), whereas the powders extracted by other solvents displayed different degrees of damage. The *X. sorbifolia* powder treated by water displayed many strip folds without crack (Figure 2b). The powder treated by ethanol and TPMBr-La showed visible pores and cracks, as shown in Figure 2c,d, respectively. Compared to the ethanol treatment, the TPMBr-La treatment showed more severe and pronounced damage. The damage was caused by the dissolution of cellulose and broken lignin after extraction with DES, which may be more conducive to the infiltration of DESs and the overflow of bioactive substances. This finding may account for the higher extraction efficiency of the DESs as compared to the organic solvent, which is consistent with the finding of polydatin extraction by DESs [34].

### 3.3. Box-Behnken Design (BBD) Optimization

To assess the influence of four variables (extraction time, extraction temperature, liquid-solid ratio, and water content in the DESs) on the extraction efficiency of TPMBr-La from *X. sorbifolia* husks, a Box-Behnken design was conducted, and the levels of the factors are presented in Table 2. The experimental orders, levels of variables, and response values are shown in the Appendix A. The quadratic regression equation for the extraction rate of total saponins (Y) and variables (A, B, C, and D) is as follows:Y = 72.16 − 1.82A + 2.92B + 0.83C − 0.62D − 0.29AB + 0.83AC + 1.71AD − 0.29BC − 0.058BD + 0.21CD − 1.27A^2^ − 3.57B^2^ − 1.43C^2^ − 3.25D^2^
(10)

The analysis of variance (ANOVA) for the regression model is displayed in Table 3. The *p*-value of the model was 0.0003 (*p* < 0.001), demonstrating that the model level was extremely significant. The coefficient of determination (R^2^) was 0.9616, and the adjusted R-square (Adj R^2^) was 0.8849, indicating that the experimental and predicted values correlate well. In addition, the *p*-value of the lack-of-fit value was not significant (0.9541), which showed that unknown factors had little interference with the test results. It can also be seen from the analysis of variance that the primary term, secondary term, and the interaction term of each factor had different effects on the extraction process. The impact degree of the independent variables was in the order of B > A > C > D. It was clear that the extraction temperature exhibited an extremely significant effect on the extraction rate (*p* < 0.0001). The *p*-values of the quadratic term coefficients A^2,^ B^2^, C^2^, and D^2^ were less than 0.05, indicating that these factors had a significant impact on the extraction rate. However, the cross-product coefficients (AB, AC, BC, BD, and CD) were not significant (*p* > 0.05).

The 3D response plots illustrated the interaction effects between the variables (Figure 3). The curve in the extraction temperature was the steepest, indicating the greatest effect on the extraction rate, and the curve of the water content was the flattest, indicating that the influence was the lowest. This conclusion is also consistent with the results of the analysis of variance. Figure 3a is used as an example of the analysis of the mutual effect of the extraction time and temperature on the extraction rate of total saponins. The increase in temperature could improve the extraction rate of total saponins, but a further increase in temperature led to a decrease in the extraction rate. This may be due to the increase in temperature, which decreased the viscosity of the deep eutectic solvents and improved the mass transfer efficiency. When the temperature is too high, the extracted saponins may be degraded, resulting in a reduction in the extraction efficiency. The highest point can be seen in the graph, suggesting that the extraction rate had the highest value. The optimum TPMBr-La conditions for the extraction of total saponins from *X. sorbifolia* were as follows: extraction time of 28 min, extraction temperature of 78 °C, liquid-solid ratios of 26 mL/g, and water content of 35%. The actual extraction rate under optimal conditions was 72.11 ± 0.61 mg Re/g dw, closely matching the predicted value of 73.06 mg Re/g dw. It is suggested that this model can simulate and predict the actual extraction of bioactive saponins from *X. sorbifolia*.

### 3.4. Recovery of Total Saponins and Kinetic Analysis of Adsorption Process

#### 3.4.1. Screening of Macroporous Resins

For the efficient recovery of total saponins from TPMBr-La extract solution, we investigated the adsorption and desorption rates of NKA-9, DM301, D101, HPD100, and HP20 macroporous resins. As described in Figure 4, D101, DM301, and NKA-9 macroporous resin displayed a superior adsorption rate (91.41%, 89.62%, 89.34%, respectively), indicating that the total saponin in the TPMBr-La extract can be adsorbed well. In the process of desorption, the D101 macroporous resin showed a better desorption rate (77.74%) for total saponins. The reason that the desorption rate is not particularly high may be attributed to the water-soluble and nonvolatile properties of DESs. Therefore, D101 resin was used to recover total saponins from the TPMBr-La extract solution.

#### 3.4.2. Kinetic Analysis of the Adsorption Process

To better understand the mechanisms of saponin adsorption on D101 macroporous resin, a kinetic analysis was conducted. The static adsorption kinetics of D101 macroporous resin on total saponins at 25 °C is shown in Figure 5a. For the first 60 min, the adsorption capacity q_t_ significantly increased as the time continued to increase. After 60 min, the adsorption rate of the resin decreased, and the adsorption capacity increased slowly as the adsorption time increased. The equilibrium adsorption qt was the highest at 10.89 mg/g when the duration was 360 min.

The adsorption process was analyzed by fitting the intra-particle diffusion, pseudo-first-order, and pseudo-second-order kinetic models. The three kinetic models fitted at 25 °C for total saponins on D101 resin are shown in Figure 5b–d. The kinetic equations of total saponin adsorption by the D101 resin and their related parameters are displayed in Table 4. C > 0 in the kinetic equation for intra-particle diffusion, suggesting that the adsorption rate of saponins partly depends on intra-particle diffusion and partly on the boundary layer diffusion [35]. The calculated values of the maximum adsorption capacity (q_e_ calc) of the pseudo-first-order and pseudo-second-order models were 6.7201 and 11.5340 mg/g, respectively. The q_e_ calc of the pseudo-second-order kinetic model was closer to the experimental q_e_ value (10.90 mg/g), and the R^2^ (0.9991) was the highest; it seemed that the adsorption process of the D101 macroporous resin was fitted to the pseudo-second-order kinetic model.

### 3.5. Identification of Barrigenol-Like Saponins in the TPMBr-La Extracts

Triterpenoids saponins are the major components in *X. sorbifolia* husks, and their backbones include the oleanane-type, lupine-type, glyseian-type, cycloatene-type, and lanolin-type [15]. The barrigenol-like triterpenoid saponins are multiple hydroxylated oleanane-type [36]. It is reported that barrigenol-like triterpenoid saponins are not only abundant in the husks of *X. sorbifolia,* but also the main responsible active components.

To find the presence of barrigenol-like triterpenoid saponins that were recovered from the TPMBr-La extracts, we performed an HPLC-ESI-MS analysis in the positive ion mode to identify the chemical compositions. The total ion chromatogram (TIC) of the saponins is shown in Figure 6. Seven types of triterpenoid saponins were identified, and their chemical structures are presented in Figure 7.

As an example, xanthoceracide (6) was identified based on ionic fragments. The precursor ion *m*/*z* 1141.57 [M + H]^+^ and 1163.57 [M + Na]^+^ in the ESI(+) mode were given, and the molecular formula could be inferred to as C_57_H_88_O_23._ The cleavage behavior of the triterpenoid saponins in the positive mode of the ESI source was highly similar. Basically, glycosides were gradually lost at first to obtain the aglycone excimer ion peaks, such as *m*/*z* 693.84 and 493.89. Moreover, the successive loss of H_2_O and angeloyl occurred in order to obtain a series of glycoside fragments, such as *m*/*z* 593.28 and 342.93. Therefore, based on similar cleavage patterns and reports in the literature [37,38,39,40], other six triterpenoids saponins were identified as 16-O-acetyl-21-O-*α*-L-rhamnopyranosyl-*β*-barringtogenol C (1), 3-O-β-D-glucopyranosyl, 28-O-[*α*-L-rhamnol (1→2)]-*β*-D-gluc-opyranosyl-16-deoxybarringtogenol C (2), 3-O-[*β*-D-glucopyranosyl (1→6)] (3′-O-angeloyl)-*β*-D-glucopyranosyl-28-O-[*α*-L-rhamnosyl(1→2)]-*β*-D-glucopyranosyl-16-deoxybarringtogenol C (3), 3-O-(3-O-angeloyl-4-O-acetyl-6-O-*β*-D-glucopyranosyl)-*β*-D-glucopyranosyl-28-O-(2-*α*-L-rhamnopyranosyl-6-O-*β*-Dglucopyranosyl)-*β*-D-glucopyranosyl-16-deoxybarringtogenol C (4), xanifoliaY_7_ (5), and xanifolia Y_2_ (7). The concrete substances identified from the extracts are listed in Appendix A. The identification of and research into barrigenol-like triterpenoids saponins in *X. sorbifolia* husks are beneficial to the further development and utilization of the saponin.

### 3.6. Evaluation of the Antioxidant Activity of the TPMBr-La Extracts

The saponins contained in the recovered extract of the ethanol from the husks of *X. sorbifolia* have been reported to have antioxidant activity [41]. Nevertheless, the impact of different extraction methods on the antioxidant activity of saponins is not clear. Thus, we evaluated the antioxidant properties of saponins extracted from *X. sorbifolia* using different methods (TPMBr-La, ethanol, and water) through a DPPH and ABTS free radical scavenging ability tests. Vitamin C (Vc) was considered as a positive control.

As shown in Figure 8, the DPPH and ABTS free radical scavenging activity of TPMBr-La extracts displayed dose-dependence. When the concentration of the TPMBr-La extract was in the range of 25~100 μg/mL, the DPPH scavenging ability enhanced with the increasing concentration. When the concentration of the extract was 100~800 μg/mL, its scavenging ability no longer improved and tended to remain stable. The DPPH free-radical scavenging activity of the TPMBr-La extracts at 100 μg/mL exhibited the highest DPPH inhibition (92.30 ± 1.10%), significantly higher than that of ethanol extraction (26.74 ± 1.00%). As can be seen from Table 5, the TPMBr-La extracts gave the lowest DPPH inhibition IC_50_ value of 36.54 ± 0.46 μg/mL (*p* < 0.001) compared to the ethanol extracts, which had the IC_50_ value of 215.35 ± 6.90 μg/mL (*p* < 0.05), while Vc demonstrated an intermediate DPPH inhibition with the IC_50_ value of 59.43 ± 0.53 μg/mL (*p* < 0.001). The ABTS free-radical scavenging rate of the TPMBr-La extracts (92.20 ± 0.30%) at 1200 μg/mL was significantly higher than that of ethanol extracts (69.89 ± 0.07%). ABTS inhibition IC_50_ value of the TPMBr-La extracts was 541.13 ± 0.03 μg/mL (*p* < 0.001), which was lower than the IC_50_ value of the ethanol extracts 855.25 ± 1.66 μg/mL (*p* < 0.001). In sum, the TPMBr-La extracts exhibited superior antioxidant activity, which may be due to the high content of saponins in the DESs solvent-extraction and the improved bioavailability through enhanced antioxidant membrane transport [42].

## 4. Conclusions

The husks of *X. sorbifolia*, as bulky by-products of industrial production, have brought a severe burden to the environment and caused an enormous waste of resources. Additionally, *X. sorbifolia* husks are known for their abundant triterpenoid saponins, particularly barrigenol-like triterpenoid saponins resources, which are responsible for a range of pharmacological bioactivities. For the high-value utilization of *X. sorbifolia* husks, we attempted to design eco-friendly and saponins-oriented DESs to extract the bioactive saponins from *X. sorbifolia* husks. TPMBr-La, as an efficient and selective solvent, was obtained from the fourteen prepared acidic DESs. The saponins’ extraction efficiency was up to 135% higher than when using 70% ethanol. The extraction mechanism showed that severe damage to the treated plant powders may be more conducive to the infiltration of DESs and the overflow of bioactive substances, to give rise to efficient extraction. The extraction conditions were optimized by Box-Behnken design RSM, and a maximum extraction rate of 72.11 ± 0.61 mg Re/g dw was obtained under the optimized extraction varies. The resulting D101 exhibited good adsorption and desorption abilities and its adsorption kinetic was compatible with the pseudo-second-order kinetic equation. The recovered TPMBr-La extracts exhibited significantly superior efficacy in scavenging the DPPH and ABTS radicals compared with the use of ethanol, indicating the probable enrichment effect of DESs on saponins. In summary, this study has demonstrated that acidic DESs are efficient and selective solvents for the high-value utilization of *X. sorbifolia* husks. Our findings also contribute to a new approach to the further application of DESs in the recycling of by-products.

## Figures and Tables

**Figure 1 antioxidants-11-00736-f001:**
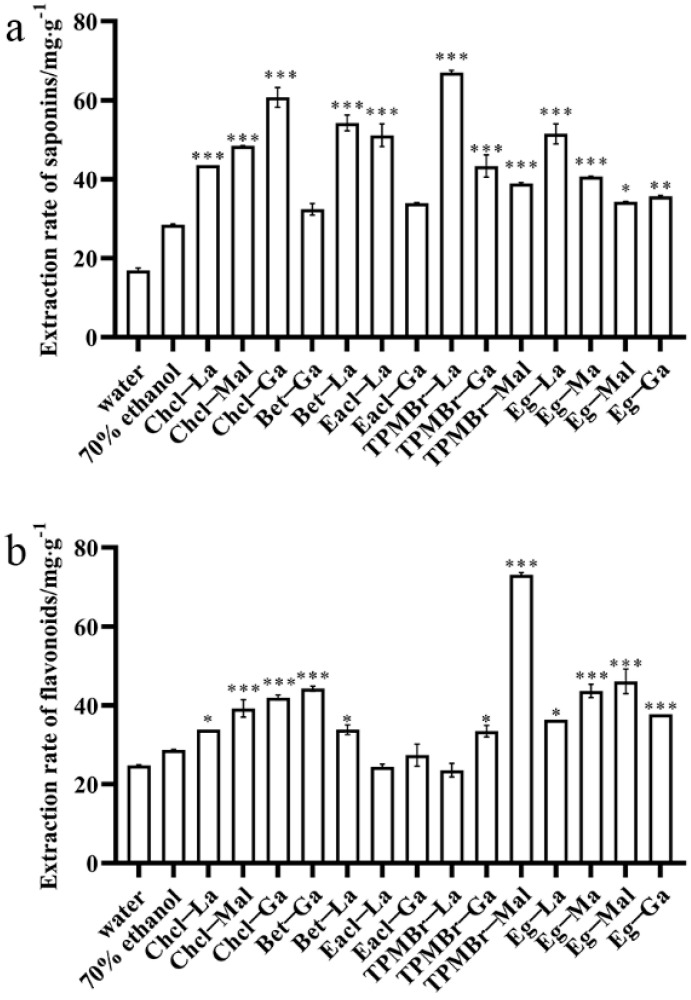
The extraction rate of total saponins and total flavonoids from *X. sorbifolia* husks using different solvents: (**a**) the extraction rate of total saponins; (**b**) the extraction rate of total flavonoids. There is a significant difference when * *p* < 0.05, ** *p* < 0.01 and *** *p* < 0.001 versus 70% ethanol.

**Figure 2 antioxidants-11-00736-f002:**
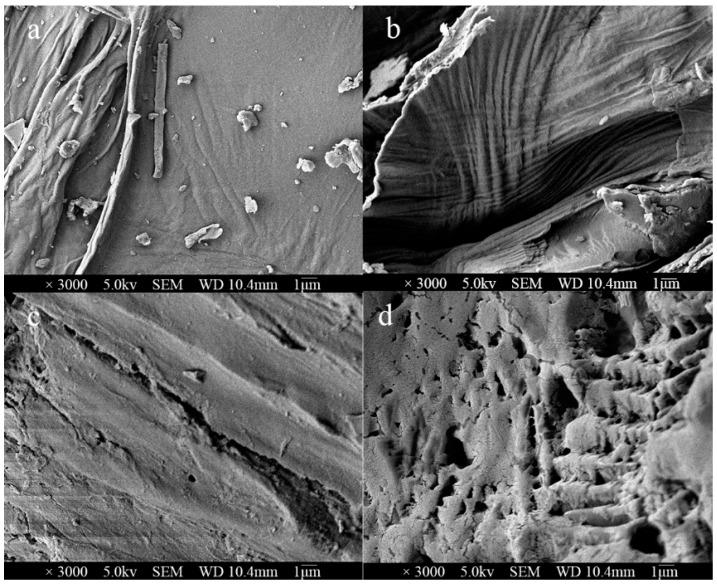
The field emission scanning electron microscope images of *X. sorbifolia* powder: (**a**) *X. sorbifolia* crude drug powder; (**b**) *X. sorbifolia* powder extracted by water; (**c**) *X. sorbifolia* powder extracted by ethanol; (**d**) *X. sorbifolia* powder extracted by TPMBr-La.

**Figure 3 antioxidants-11-00736-f003:**
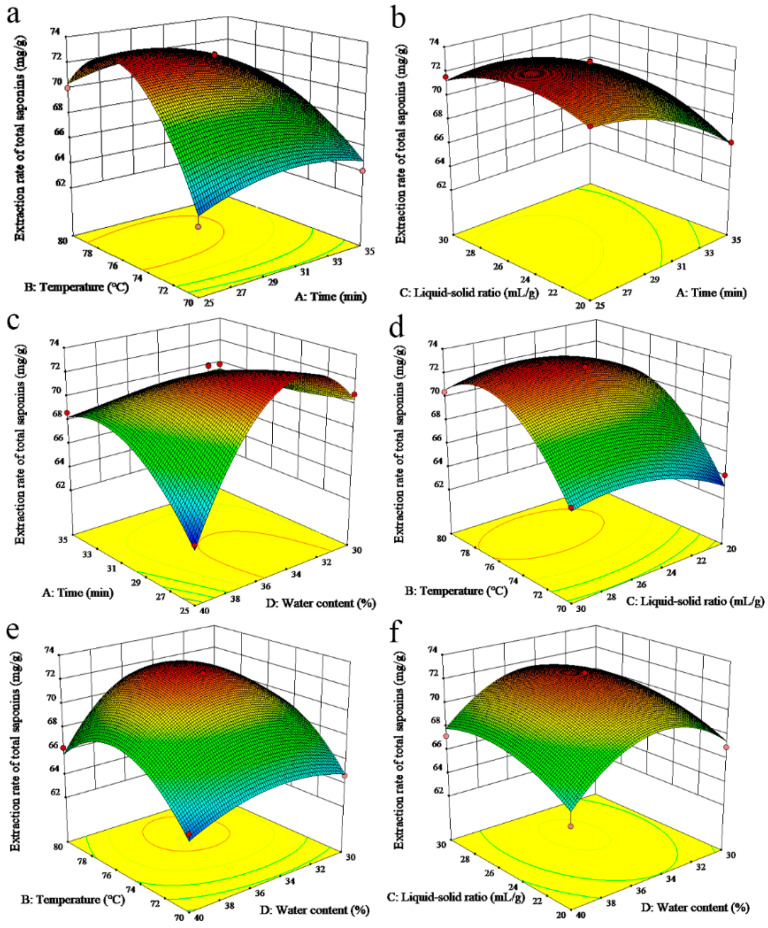
Comprehensive score of the response surface of the total saponins extraction rate. (**a**) The interaction between extraction time and temperature; (**b**) the interaction between liquid–solid ratio and extraction time; (**c**) the interaction between extraction time and water content; (**d**) the interaction between liquid–solid ratio and extraction temperature; (**e**) the interaction between water content and extraction temperature; (**f**) the interaction between water content and liquid–solid ratio.

**Figure 4 antioxidants-11-00736-f004:**
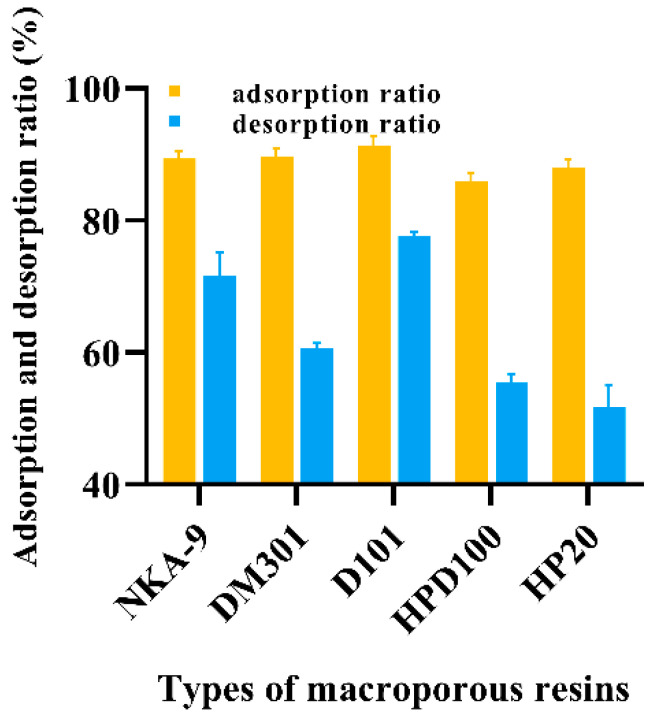
Adsorption and desorption rates of different kinds of macroporous resins.

**Figure 5 antioxidants-11-00736-f005:**
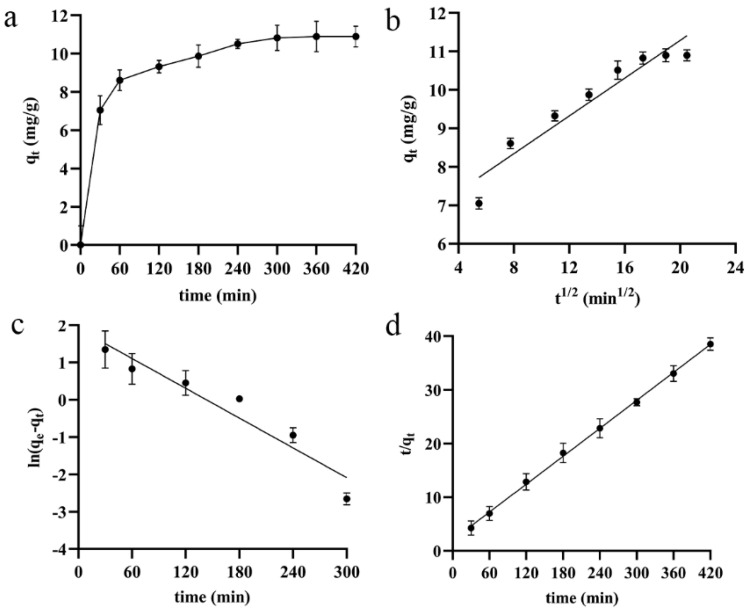
D101 static adsorption kinetic curve of macroporous resin (**a**); intra-particle diffusion (**b**); pseudo-first-order (**c**); pseudo-second-order (**d**).

**Figure 6 antioxidants-11-00736-f006:**
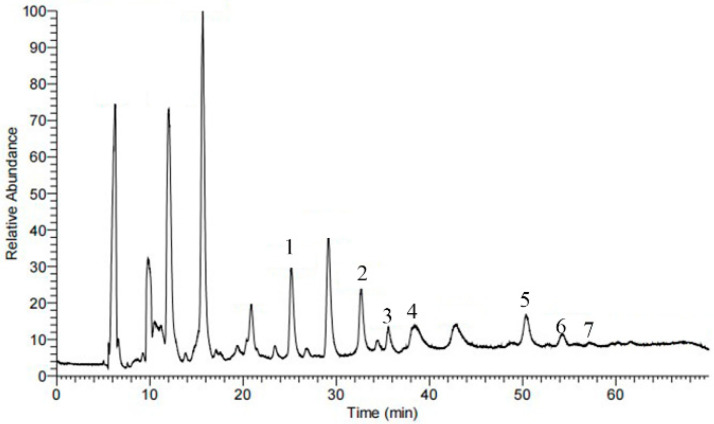
Total ion chromatogram (TIC) 200–2000 *m*/*z* of the TPMBr-La-extract saponins of *X. sorbifolia*.

**Figure 7 antioxidants-11-00736-f007:**
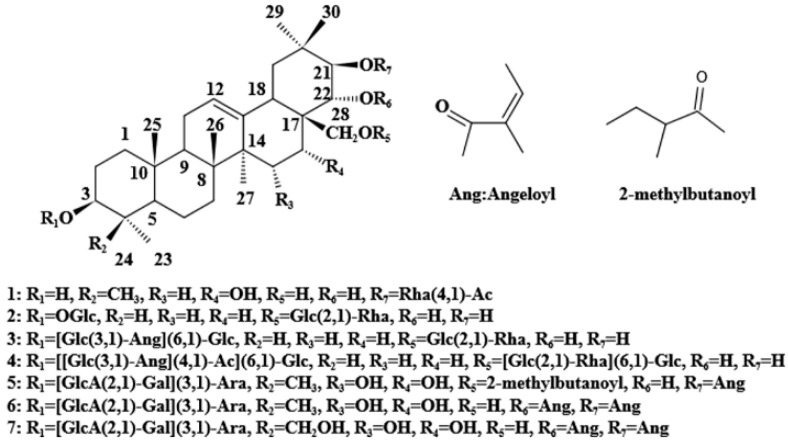
Chemical structure of the saponins identified from *X. sorbifolia* husks.

**Figure 8 antioxidants-11-00736-f008:**
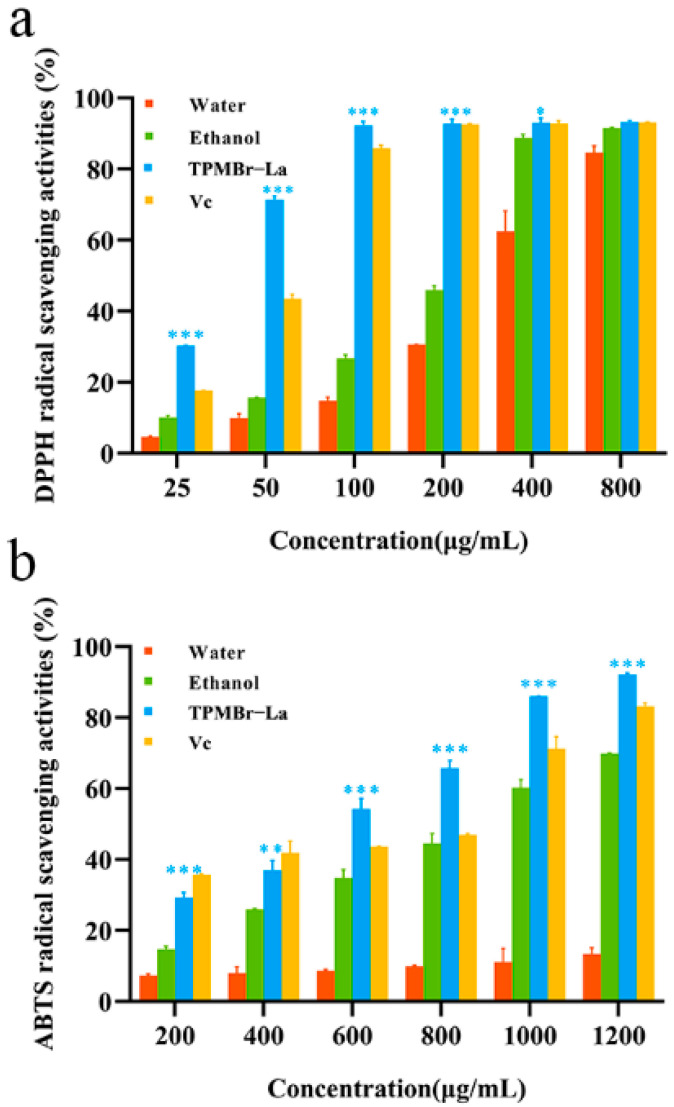
Determination of the DPPH and ABTS radical scavenging capacity of bioactive saponins recovered from water, ethanol, and TPMBr-La. There is a significant difference when * *p* < 0.05, ** *p* < 0.01, and *** *p <* 0.001 versus ethanol.

**Table 1 antioxidants-11-00736-t001:** Composition and solvent code for the studied DESs.

No.	Solvent Code	Hydrogen Bond Acceptors (HBAs)	Hydrogen Bond Donors (HBDs)	Molar Ratio
1	Chcl-La	Choline chloride	Lactic acid	1:2
2	Chcl-Mal	Choline chloride	Malonic acid	1:2
3	Chcl-Ga	Choline chloride	Glycolic acid	1:2
4	Bet-Ga	Betaine	Glycolic acid	1:2
5	Bet-La	Betaine	Lactic acid	1:2
6	Eacl-La	Ethylamine hydrochloride	Lactic acid	1:2
7	Eacl-Ga	Ethylamine hydrochloride	Glycolic acid	1:2
8	TPMBr-La	Tetrapropylammonium bromide	Lactic acid	1:2
9	TPMBr-Ga	Tetrapropylammonium bromide	Glycolic acid	1:2
10	TPMBr-Mal	Tetrapropylammonium bromide	Malonic acid	1:2
11	Eg-La	Ethylene glycol	Lactic acid	1:2
12	Eg-Ma	Ethylene glycol	Malic acid	1:2
13	Eg-Mal	Ethylene glycol	Malonic acid	1:2
14	Eg-Ga	Ethylene glycol	Glycolic acid	1:2

**Table 2 antioxidants-11-00736-t002:** Independent factors and their levels.

Independent Factor	Levels		
	−1	0	1
A Extraction time (min)	25	30	35
B Extraction temperature (°C)	70	75	80
C Liquid-solid ratio (mL/g)	20	25	30
D water content (%)	25	30	35

**Table 3 antioxidants-11-00736-t003:** The ANOVA of the regression model.

Variables	Sum of Squares	df	Mean Square	F-Value	*p*-Value
Model	282.84	18	15.71	12.547	0.0003
A-Extraction time	13.25	1	13.25	10.57	0.0100
B-Extraction temperature	67.98	1	67.98	54.23	<0.0001
C-Liquid-solid ratio	8.23	1	8.23	6.57	0.0305
D-Water content	3.11	1	3.11	2.48	0.1495
AB	0.34	1	0.34	0.27	0.6169
Model	282.84	18	15.71	12.547	0.0003
A-Extraction time	13.25	1	13.25	10.57	0.0100
B-Extraction temperature	67.98	1	67.98	54.23	<0.0001
C-Liquid-solid ratio	8.23	1	8.23	6.57	0.0305
D-Water content	3.11	1	3.11	2.48	0.1495
AB	0.34	1	0.34	0.27	0.6169
AC	2.72	1	2.72	2.17	0.1746
AD	11.73	1	11.73	9.36	0.0136
BC	0.35	1	0.35	0.28	0.6110
BD	0.013	1	0.013	0.011	0.9204
CD	0.17	1	0.17	0.13	0.7227
A^2^	11.08	1	11.08	8.84	0.0156
B^2^	87.33	1	87.33	69.67	<0.0001
C^2^	14.02	1	14.02	11.19	0.0086
D^2^	72.21	1	72.21	57.60	<0.0001
Residual	11.28	9	1.25		
Lack of Fit	3.24	6	0.54	0.20	0.9541
R^2^	0.9616				
Adj R^2^	0.8849				

**Table 4 antioxidants-11-00736-t004:** Kinetic equations for saponins adsorption by the D101 resin.

Kinetic Model	Equations	q_e_ exp.	qe calc.	K	R^2^
Intra-particle diffusion	qe = 0.2454t1/2 + 6.3797	10.90	11.0358	0.2454	0.9168
pseudo-first-order	ln (qe−qt) = −0.0133t + 1.9051	10.90	6.7201	0.0133	0.9211
pseudo-second-order	$\frac{t}{q_{t}}$ = 0.0867t + 2.0116	10.90	11.5340	0.0037	0.9991

**Table 5 antioxidants-11-00736-t005:** The IC_50_ value of DPPH and ABTS free-radical scavenging activity.

Samples	IC_50_ (μg/mL)	
	DPPH	ABTS
Water	321.79 ± 6.90	>1200
Ethanol	215.35 ± 5.43	855.25 ± 1.66
TPMBr-La	36.54 ± 0.46	541.13 ± 0.03
Vc	59.43 ± 0.53	620.14 ± 3.46

## Data Availability

The data presented in this study are available in this manuscript.

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
