# Peer review of "Oriented Deep Eutectic Solvents as Efficient Approach for Selective Extraction of Bioactive Saponins from Husks of *Xanthoceras sorbifolia* Bunge"

_antioxidants, 2022, doi:10.3390/antiox11040736_

Round 1

Reviewer 1 Report

This work is about the potential of DES as designer solvents to recover saponins from industrial by-products, X. sorbifolia in particular.

This is a hot topic regarding the green chemistry goals. Furthermore, compared with other natural added value molecules (eg. Phenolics, pigments, cellulose…), this manuscript stands out by exploring a less usual (but real important) class of compounds: saponins.

However, one clear bottleneck is the English spelling and grammar. This manuscript has potential, but it needs to be carefully revised from title to conclusion sections.

Title: the authors are repeating “saponins” twice. I suggest simplifying it.

Abstract: English style must be revised. This section is too long, please redact it with a focus on the main objectives and achievements.

Introduction

Line 46-59: please, add references. The amounts of waste are based on which source?

Line 69: “ether et al.”? please correct this sentence properly, adding the needed ref.

Line 85: “DESs synthesized”. Please revise the concept of DES, they are a mixture of components and not a result of a chemical reaction. Therefore, DES are prepared/composed of/a mixture of… Correct this phrase properly.

Line 86: Ref 17 – there is a massive number of works measuring the total phenolic content of extracts obtained by DES as solvent. Why use this ref instead of a recent revision about this topic? It makes any sense. Please revise this carefully.

Line 87: But there are other works about DES and saponins, which should be included in this section instead of a random ref about phenolic contents in another matrix.

Line 118: again, DES are not a result of a synthesis process. Please revise all the manuscript concerning this topic. I encourage the authors to see the following review about DES: https://doi.org/10.1007/s10953-018-0793-1

Table 1: (1) what was the criteria to select these starting materials? (2) Also, why the authors do not test Bet-Mal nor Eacl-Mal? Adding 30% of water they will be (probably) liquids too. (3) why 1:2 molar ratio for all cases? DES composed of acids usually had stoichiometry according to the number of carboxylic groups.

Section 2.3: please be clear about the solvents used besides DES. Also, the 30% of added water was w/w, v/v, w/v…?

Figure 1: No section was presented regarding statistical analysis. Please add it to section 2.

Line 255-256: why it was so obvious? The plant material has (in usual) higher amounts of saponins than other biomolecules? Acidic DES, especially the ChCl-acidic based ones, is excellent for extracting phenolics from biomass (https://doi.org/10.1016/j.indcrop.2018.02.029).

Line 269 and 271: “ways”?!? please select another word; “solvents” seems appropriate. Revise all the manuscript, please.

Line 273: “others” (=)“the extracted samples”? Please, take care when describing the results.

Line 278-280: (…) DES than those others. Do the authors have a clue why? Can you please add an explanation to this capacity of acidic DES better extract the target molecules in comparison with conventional solvents such as ethanol?

Line 280-281: Why compare DES performance with IL’s? It will be more fruitful a discussion under the same topic (as suggested in the question above). You are not comparing equivalent DES and IL, therefore this discussion makes no sense.

I noticed that the scientific name of the plant is not in the italic form in figure captions. Please correct this.

Figure 3: Are the authors sure that data analysis was carried out by using Design Expert 8.0.5? These plots seemed to be obtained by Statistica…

Line 388-405: I suggest adding a table to summarize these results. Important columns will be RT, ID, molecular ion, fragmentation pattern, quantification and Refs supporting the ID.

Line 394: “literature reports”, which ones? Add refs, please.

Section 3.6: It would be great if the authors provided IC50 values. This is extremely important when comparing results, allowing future researchers to cite your work. You only can’t determine it for the water extract at ABTS assay, but you always can express the result as (>1200 ug/mL).

Reviewer 2 Report

I have no negative comments. The manuscript is complex, with innovative and future results. The use of these solvents - DES - seems to be a selective method, with a higher yield than the classical methods and with many advantages.

Author Response

Dear Editors and Reviewers:

Thank you for your letter and for the reviewers’ comments concerning our manuscript entitled “Saponins-oriented deep eutectic solvents as efficient approach for selective extraction of bioactive saponins from shells of Xanthoceras sorbifolia Bunge (ID: Antioxidants-1644102). Those comments are all valuable and very helpful for revising and improving our paper, as well as the important guiding significance to our research.

I wish you every success in your work.

Reviewer 3 Report

The article is up-to date. It concerns problems concerning the usage of by-products and the results presented are novel, of high potential to be implemented. However, the text needs corrections and additional information.

The use of the term shell is highly misleading. From the title, one might suppose that the text concern the shells of some shellfish, and not part of the fruit collected from trees. The botanical nomenclature should be introduced in to the article to describe carefully the raw material collected and used for the experiment. This should be specified already in the introduction and proper name should be used all over the text.

Line43-45: From the text we don’t know what type of raw materials are used for pharmaceuticals production (seeds, leaves, flowers?).

Line 54-55: “feed et al.” - it was supposed to be a quotation? Incomprehensible.

Line 60: “which” is not needed here.

Line 63/64- “and other disease” is redundant her (says nothing)

Line 83: what “Acid compounds”? – unclear

Line 86: “total phenolic” content? (line 85-87- unclear)

The last paragraph of the introduction usually presents the aim and research hypothesis, and not the results obtained. In the work presented for review, it looks like a summary. This part should be verified in accordance with the Journal requirements.

“Introduction: The introduction should briefly place the study in a broad context and highlight why it is important. It should define the purpose of the work and its significance, including specific hypotheses being tested. The current state of the research field should be reviewed carefully and key publications cited. Please highlight controversial and diverging hypotheses when necessary. Finally, briefly mention the main aim of the work and highlight the main conclusions. Keep the introduction comprehensible to scientists working outside the topic of the paper.”

Line 103: What is “standardized” garden?

Line102- 107- this is completely unclear. What species was the object of the study: Xanthoceras sorbifolia or Populus euramericana? What Populus euramericana has to do with this experiment? What was the raw material used for the experiment - the parts of fruits or the leaves? When and how the raw materials were collected? How it was dried? These are the basic information to assess the reliability of the research results obtained.

Table 1. For better readability, I suggest to provide full names of abbreviations (HBAs, BHBDs) under the table.

Round 2

Reviewer 1 Report

The authors carefully revised the manuscript according to each comment. It was remarkable the efforts made!

The results of Table 5 (antioxidant properties) should be presented as mean values and SD. Of course, the statistical analysis also needs to be addressed.

Author Response

Thank you for your letter and for the reviewers’ comments concerning our manuscript entitled “Saponins-oriented deep eutectic solvents as efficient approach for selective extraction of bioactive saponins from shells of Xanthoceras sorbifolia Bunge (ID: Antioxidants-1644102). Those comments are all valuable and very helpful for revising and improving our paper, as well as the important guiding significance to our research.

We tried our best to improve the manuscript and made some changes to the manuscript. These changes will not influence the content and framework of the paper. And here we marked in blue in the revised paper. We appreciate for Editors and Reviewers warm work earnestly and hope that the correction will meet with approval. Once again, thank all of you very much for your comments and suggestions.
